# Production and Characterization of a Novel Exopolysaccharide from *Ramlibacter tataouinensis*

**DOI:** 10.3390/molecules27217172

**Published:** 2022-10-24

**Authors:** Desislava Jivkova, Ganesan Sathiyanarayanan, Mourad Harir, Norbert Hertkorn, Philippe Schmitt-Kopplin, Ghislain Sanhaji, Sylvain Fochesato, Catherine Berthomieu, Alain Heyraud, Wafa Achouak, Catherine Santaella, Thierry Heulin

**Affiliations:** 1CEA, CNRS, Laboratory for Microbial Ecology (LEMiRE), Aix Marseille University, UMR7265 BIAM, F-13108 Saint-Paul-lez-Durance, France; 2Research Unit Analytical BioGeoChemistry (BGC), Helmholtz Munich, Ingolstädter Landstr. 1, D-85764 Neuherberg, Germany; 3Analytical Food Chemistry, Technical University Munich, Maximus-von-Imhof-Forum 2, Weihenstephan, 85354 Freising, Germany; 4White Biotechnology Department, ARD, F-51110 Pomacle, France; 5CNRS, Aix Marseille University, FR 3098 ECCOREV, F-13545 Aix-en-Provence, France; 6CEA, CNRS, IPM, Aix Marseille University, UMR7265 BIAM, F-13108 Saint-Paul-lez-Durance, France; 7CNRS, CERMAV, 38000 Grenoble, France

**Keywords:** *Ramlibacter tataouinensis*, exopolysaccharide, biosynthesis, optimization, bioreactor, characterization, GC, NMR, FT-ICR-MS, FTIR

## Abstract

The current study examines the desiccation-resistant *Ramlibacter tataouinensis* TTB310^T^ as a model organism for the production of novel exopolysaccharides and their structural features. This bacterium is able to produce dividing forms of cysts which synthesize cell-bound exopolysaccharide. Initial experiments were conducted on the enrichment of cyst biomass for exopolysaccharide production under batch-fed conditions in a pilot-scale bioreactor, with lactate as the source of carbon and energy. The optimized medium produced significant quantities of exopolysaccharide in a single growth phase, since the production of exopolysaccharide took place during the division of the cysts. The exopolysaccharide layer was extracted from the cysts using a modified trichloroacetic acid method. The biochemical characterization of purified exopolysaccharide was performed by gas chromatography, ultrahigh-resolution mass spectrometry, nuclear magnetic resonance, and Fourier-transform infrared spectrometry. The repeating unit of exopolysaccharide was a decasaccharide consisting of ribose, glucose, rhamnose, galactose, mannose, and glucuronic acid with the ratio 3:2:2:1:1:1, and additional substituents such as acetyl, succinyl, and methyl moieties were also observed as a part of the exopolysaccharide structure. This study contributes to a fundamental understanding of the novel structural features of exopolysaccharide from a dividing form of cysts, and, further, results can be used to study its rheological properties for various industrial applications.

## 1. Introduction

The high demand for natural polysaccharides in different industrial sectors has focused attention on microbial exopolysaccharides (EPSs). EPSs are long-chain biopolymers composed of branched and repeating units of monosaccharides, which are mainly produced by various microorganisms when excess carbon is available in the growth environment and in case of environmental stresses [1]. EPSs are synthesized by bacteria and secreted out of the cells as a capsule or slime, which is loosely associated to the bacterial cell surface [2]. These EPS molecules are mainly responsible for biofilm formation and are also involved in, e.g., cell defense mechanisms to cope with osmotic, saline, and low/high temperature stress, desiccation, antibiotics, and antibodies [3]. Due to their excellent physicochemical and material properties, EPSs are broadly used in, e.g., the food, pharmaceutical, petroleum, cosmetic, and medical industries as emulsifiers, biosurfactants, gelling agents, ion exchange resins, viscofiers, bioflocculants, stabilizers, and drug carries [4]. In addition, EPS molecules have also been screened for their antibacterial, antitumor, antiviral, and anti-inflammatory properties [5]. EPSs are currently being exploited as additional commercial products such as bacterial cellulose, succinoglycan, curdlan, dextran, pullulan, xanthan, gellan, alginates, and hyaluronic acid from different bacterial and fungal strains [4].

The biosynthesis of EPS is a special metabolic process in which diverse polysaccharides with distinctive features can be synthesized by unrelated strains of the same species [6]. Nowadays, there is considerable interest amongst scientists toward finding novel EPS structures with unique properties. As a result of screening efforts, many bacteria from various extreme environments have been sourced for novel EPSs that can be exploited for different biotechnological applications [7]. However, arid and semi-arid environments and their microorganisms have scarcely been subjected to the biosynthesis and characterization of novel EPS structures. One such example is *Ramlibacter tataouinensis* TTB310, a slow-growing betaproteobacterium isolated from sand particles from a semi-arid region of South Tunisia (Tataouine) [8]. This bacterium possesses the unusual characteristic of coexistence in two morphological forms that are able to divide into spherical cysts and rod-shaped cells [8,9,10]. This unique cell cycle allows this strain to adapt to the desert environment, and this strain is tolerant to desiccation due the formation of cysts [10,11]. It is believed that the major reason for cysts’ long-term tolerance to desiccation is the fact that they are embedded within a thick layer of EPS, which is not present in the rods [8,11]. The complete genome sequence is available for this strain of *R. tataouinensis*, which confirms the presence of 40 chromosomal glycosyl-transferases (GTs), and these GTs are supposedly responsible for the formation of α-, β-linked D-gluco configurations in the glycosidic bonds of the EPSs [11].

Transmission electron microscopy studies have shown that the EPSs of cysts, appearing as a dark halo, were arranged in a thin and compact layer firmly attached to the cellular surface [8,9]. These features are different from many of the commercialized bacterial EPSs like cellulose and xanthan, produced in high quantity and excreted in the culture medium [6,12]. Another peculiarity is that, unlike the bacteria used industrially (e.g., *Xanthomonas*, *Rhizobium*) [13,14], the EPS of *R. tataouinensis* is produced during the logarithmic growth phase with cyst divisions rather than only during the stationary phase [11]. Therefore, the optimization of the culture medium for better EPS production by *R. tataouinensis* and the extraction procedure should be conceived differently compared to classical industrially produced EPS. To our knowledge, however, no information is available either on the biosynthesis of the EPSs from *R. tataouinensis* or on their structure. Therefore, this study aimed to optimize cyst production for the biosynthesis of EPS by *R. tataouinensis* TTB310^T^ (=DSM 14655T =ATCC BAA-407^T^ =LMG 21543^T^) and to determine the structural information of synthesized EPS including monosaccharide composition, using FT-ICR-MS, NMR, and FTIR.

## 2. Results and Discussion

### 2.1. Optimization of Cyst Formation in Batch Fermentation

Usually, optimization of EPS production is performed by two-step processes such as optimization of cell growth and EPS production from biomass while increasing the carbon/nitrogen (C/N) ratio [4]. The main difficulty we have had to face for the optimization of EPS production by *R. tataouinensis* TTB310 was related to the fact that only cysts of strain TTB310 were able to produce EPS (8,10,11). In this study, strain TTB310 grown in TSB 1/10 also exhibited both cysts and rods (Appendix A), and it was difficult to induce the differentiation of dividing rods into cysts in order to use the two-step strategy in a controlled manner. Therefore, we decided to take advantage of one main peculiarity of this bacterium, which is the ability of cysts to divide and so to optimize simultaneously the cell division of strain TTB310 cysts and their EPS production.

In batch culture (TSB 1/10), the growth of strain TTB310 was significantly increased in a reproducible way in the presence of lactate (OD 0.19). Stimulation of growth was maximal at 10 mM lactate (10^8^ to 10^9^ cfu/mL^−1^) and remained significant at higher concentrations (20 and 40 mM). A yellow-orange color and highly mucous colonies were obtained with a high proportion of cysts when the medium was supplemented with 10 mM lactate (Appendix A). With respect to biomass, cultures of strain TTB310 in TSB 1/10 medium had an average dry biomass of about 4.8 mg from 150 mL of medium broth, whereas cultures in 10 mM lactate-supplemented medium had an average of 10.8 mg biomass. Therefore, bacterial growth was approximately two times higher in cultures supplemented with 10 mM lactate when compared to lactate-free control. Although in previous studies *R. tataouinensis* was shown to assimilate acetate, lactate, and β-hydroxybutyrate as carbon and energy sources [8,11], we did not observe such an important stimulation of growth in the presence of acetate, hydroxybutyrate, and succinate, probably because only lactate can regulate pH medium (see Section 2.2).

### 2.2. Optimization of Cyst Formation in Fed-Batch Fermentation

Two different fed-batch experiments were executed based on the pH. First, the experimental (TSB 1/10) medium was supplemented with 10 mM of lactate with an initial pH of 7.4, and the lactate was fed into the bioreactor by two consecutive additions (2.5 and 4th day of incubation). In a second experiment, pH was regulated continuously by adding lactic acid as a pH regulator and carbon source. The initial dissolved oxygen (DO) level (100%) had gradually decreased to 50% at the end of the first experiment (Appendix A). In addition, the DO level did not fall below the threshold of 20% which would trigger a cascade of regulations. Therefore, oxygenation did not seem to be the limiting factor for cyst formation and subsequent EPS synthesis during the first experiment. In contrast, during the second bioreactor culture, especially after three days of fermentation, the culture was enriched with aggregates of cysts. On the other hand, the medium became alkaline (pH 8.7) at the end of the first experiment (Appendix A). After each addition of lactic acid (final concentration: 2.2 mM), the pH dropped sharply but became alkaline again in less than 24 h (Appendix A). Two lactic acid doses did not seem to affect the growth kinetics of strain TTB310 (Appendix A). Stopping the lactic acid supply beyond the 7th day of culture resulted in an immediate rise in pH 8.4 in the second experiment (Appendix A) and also arrested the growth of strain TTB310 or even induced the bacterial cell lysis (Appendix A). This observation confirms the sensitivity of *R. tataouinensis* to alkalinity, and high pH is one of the significant limiting factors for cysts and EPS synthesis.

In terms of strain TTB310 growth, TSB 1/10 with 10 mM lactate medium without pH regulation had shown an OD of 0.16 (Appendix A). pH regulation led to a significant improvement in growth (OD 0.25) while maintaining the same nutritive conditions after 7 days of fermentation (Appendix A). When using starting medium with tryptone 1g/L and yeast extract 0.3 g/L, a drastic increase in the growth (OD 0.60) under fed-batch condition resulted, higher than that of all of the results obtained before (Appendix A). Finally, a rich starting medium (10 mM lactate, 4 g/L tryptone, 1.3 g/L yeast extract, 0.25 g/L K_2_HPO_4_, and 0.5 g/L NaCl) with continuous pH regulation produced an OD of about 0.58. At this point, strain TTB310 culture was highly enriched with cysts. Adding a supplement (tryptone 1g/L, yeast extract 0.3 g/L) on the 4th day of fermentation resulted in further enhanced growth, with OD reaching 0.76. However, an increased proportion of rods and a decrease in cysts’ aggregates was noticed during this second phase of growth. Therefore, we concluded that stopping the culture at days 3 to 4 would optimize the mass production of strain TTB310 cysts and subsequent EPS extraction for chemical characterization.

### 2.3. Extraction, Solubility, and Chemical Analysis of EPS

The modified extraction method using PBS with TCA has significantly increased the yield of EPS, since TCA particularly denatures and precipitates the proteins. This modified extraction method was devised based on literature [15,16] and the peculiarities of *R. tataouinensis* TTB310. Calcofluor white stain revealed that the EPS had detached from the surface of the cysts, because the fluorescent halo around the cysts was no longer present. Only filamentous fluorescent residues were observed under optical microscopy, confirming the EPS separation from cysts (Figure 1). Despite the elimination of TCA and lipids, the EPS of strain TTB310 remained insoluble in 0.1% final concentration in 0.1 M NaCl, even after pH adjustment, stirring, and progressive heating from 40 °C to 90 °C. However, EPS readily dissolved in DMSO. This solution was subsequently dialyzed.

Mass elemental compositions such as C, H, N, and O were determined to be 41.3 ± 2.0%, 7.4 ± 0.4%, 5.7 ± 1.3%, and 49.4 ± 4.0%, respectively (three independent samples from the same EPS extraction). These values were computed after the subtraction of the TCA mass, since it had been used for the purification of the EPS. Uronic acids and RNA traces were not detected. The total monosaccharide content in purified EPS following resuspension in DMSO showed an average of 42% attenuation in monosaccharides. Elemental analysis also detected nitrogen in the samples, possibly resulting from peptides or membrane proteins solubilized during the extraction. A second hypothesis is that the EPS of strain TTB310 carries amine or amide groups as substituents of monosaccharides or amino acids in its repeating unit. Chemical analysis indicates that the extracted EPS was composed of repeating units of a saccharide nature with possible substitutions, for example with acetyl groups.

HPLC analysis shows that only about 1.5% of lipids were present in the total EPS samples, and the HPLC profile also confirms the presence of outer membrane polar lipids. This might arise from the extraction process, since the EPS of strain TTB310 was tightly bound to the outer membranes of the bacterial cells.

### 2.4. Monosaccharide Composition and Quantification by GC Analysis

Two different GC analyses of the EPS samples of strain TTB310 were performed after hydrolysis with sulfuric acid (13 M H_2_SO_4_, then 1 M) and methanolysis. Hydrolysis with H_2_SO_4_ detected five hexoses, two of them were predominant—glucose (Glc) and galactose (Gal)—and three were minor, such as mannose (Man), xylose (Xyl), arabinose (Ara), and the pentose ribose (Rib) (Figure 2).

The presence of ribose in the EPS requires verification by other methods, since this pentose is rarely present in bacterial EPSs described in the literature [17]. From the GC analysis of EPS after hydrolysis with H_2_SO_4_, the monosaccharide composition was estimated as follows: 7.7%, 3.0%, 0.4%, 0.2%, and 0.3% representing glucose, galactose, mannose, arabinose, and xylose, respectively. The total mass percentage of all monosaccharides was about 12%. Therefore, the rest of the monosaccharides (according to elemental analysis 38.5% or 42% via colorimetric test) was not identified by this method. The difference between these values may come from a low yield of hydrolysis and derivatization (about 50%) or from incomplete identification of the constituents of the repeating unit (e.g., oxidized sugar moieties).

The GC analysis of methanolized EPS revealed the presence of ribose, glucose, galactose, and mannose. Interestingly, certain monosaccharides (e.g., arabinose and xylose) were absent (Figure 3). 

The presence of these peaks may indicate monosaccharides different from the “standard” molecules used or those linked to substituents that have not been released after methanolysis. These can be non-glycosidic molecules bearing hydroxyls, which allowed for their silylation and subsequently their volatilization. According to this second GC analysis, the total percentage of the major monosaccharides amounted to 12–14%, distributing across ribose (5.2%), glucose (4.0%), galactose (2.8%), and mannose (1.7%). The ratio of the four species (Rib:Glc:Gal:Man) from an average of two methanolyses was as follows: 1.5:1.0:0.7:0.4, or 3/2/1/1. In our hands, methanolysis was found to be a less efficient hydrolysis method for this EPS compared to sulfuric acid hydrolysis.

### 2.5. EPS Polymer Composition and Structural Evaluation by FT-ICR-MS

The ultrahigh-resolution mass spectrometry (FT-ICR-MS) of the EPS of *R. tataouinensis* TTB310 between 100 and 1500 *m*/*z* is depicted in Figure 4A. The elementary raw formulas were calculated using an in-house written script, with a tolerance of 0.5 ppm between the detected mass and the theoretical mass. The theoretical mass was computed on the basis of the gross formula of the ion, and the theoretical mass of the ion is labelled “experimental monoisotopic mass” with *m*/*z* accounting for the negative charge [18]. Figure 4 shows the visualization of a van Krevelen diagram (Figure 4B) and mass edited H/C ratios plot of 32 compounds characteristic of CHO-containing compounds (Figure 4C), corresponding only to the detected carbohydrate compounds (i.e., mono- and oligosaccharides). For each of these compounds, we searched databases (ChemSpider, PubChem) for potential structural candidates corresponding to mono- and oligosaccharides, considering that these raw formulas originated from different cases:Loss of a proton and formation of (M-H)^−^Loss of a proton, dehydration and formation of (M-H_2_O-H)^−^Loss of a proton, 0.2A fragmentation and formation of (M-C_2_H_4_O_2_-H)^−^Loss of a proton, 0.2A fragmentation, dehydration and formation of (M-C_2_H_4_O_2_-H_2_O-H)^−^

A table of 120 potential compounds was generated with possible structures generated by a Metlin *m*/*z* match search in negative mode on the basis of (M-H)^−^ and (M-H_2_O-H)^−^ ions, and a list of gross formulae corresponding to neutral mass losses (M-H_2_O) was also computed. This database was completed by adding the most characteristic motifs/substituents of monosaccharides (Appendix A) and typical mass losses corresponding to classical fragmentations of monosaccharides (Appendix A). Some fragmentation patterns revealed the type of linkage of monosaccharides (1.2, 1.3, 1.4, 1.6) [19].

In total, a table of 180 neutral mass-loss patterns was generated, and a list of 32 *m*/*z* reflecting these mass losses was used to construct a distance matrix among these *m*/*z* values. Then the list of mass losses (neutral patterns) was used to search for patterns in the distance matrix with a precision of 5 decimal places; Appendix A shows an example of such a pattern search in the distance matrix. The neutral mass loss patterns and potential structures assigned to the *m*/*z* list allowed for the reconstruction of the structure of potential fragments; an example is shown in Appendix A. On the same principle, we generated the interpretation Appendix A. The tolerance ∆/M between the experimental and theoretical mass for *m*/*z* was ±0.7 ppm. The evidence presented in Appendix A leads to the following plausible units:

(UA)-(Tri-Pentose)-(Hexose)-(Hexose-di-OAc)-(Hexose-Succ)-(Hexose-OMe-(6-Deoxyhexose)-(6-Deoxyhexose)→

This oligosaccharide contained four hexoses, three pentoses, two deoxyhexoses, one uronic acid, two acetyls, an O-methyl, and a succinyl, and its formula (C_66_H_100_O_51_) can be detected on the global spectrum as a fragment ^0^.^2^A (M-C_2_H_4_O_2_-H)^-^, C_64_H_95_O_40_)^-^ at *m*/*z* 1647.49429 (intensity 2.08 × 10^6^, theoretical mass 1647.49474, ∆/M= −0.27314 ppm). Using the indices given by fragmentation, the mass spectrum appears to represent the following plausible repeated units:

(UA)-(Tri-Pentose)-1→6-(Hexose)-1→6-(Hexose-3,4-di-OAc)-(Hexose-Succ)-(Hexose-OMe)-(6-Deoxyhexose)-(6-Deoxyhexose)→

Apart from the list of *m*/*z* selected by the ratio H/C vs O/C, the spectrum showed very intense mass peaks, corresponding to adducts (M+Cl_3_COO)^−^ and (M+HSO_4_)^−^, with characteristic isotopic patterns.

### 2.6. EPS Polymer Composition and Structural Evaluation by FT-ICR-MS

Several NMR spectra were acquired on the supernatant of a suspension after 5 h of sonication with D_2_O. Spectral analysis revealed that the anomeric protons (O_2_C**H** units) resonated generally from 4.4–5.8 ppm, with alpha protons between 4.8 and 5.8 ppm and beta protons between 4.2 and 4.8 ppm (Figure 5 and Appendix A). 

Anomeric carbon (O_2_**C**H units) resonated from 95–110 ppm, with alpha carbon resonances from 98–103 ppm and beta carbon resonances from 103–106 ppm. Hydrogen and carbon, connected by scalar coupling constants ^1^JCH in HSQC (heteronuclear single quantum coherence) and ^3^J_H1H2_ in TOCSY NMR spectra (total correlation spectroscopy), provided information about the saccharide configuration. OC**H** units in the saccharide ring generally resonated from 3.2–4.5 ppm. The methyl groups of 6-deoxy saccharides and O- and N-acetyl groups were in the range of 1.2 to 2.3 ppm. The carbon atoms in rings (O**C**H units) resonated from 62–85 ppm and appeared in the case of attached nitrogen functional groups near 50–60 ppm, and in the case of common, simple oxygenated carbon (O**C**H units) in the 65–75 ppm range. The carbon atoms involved in glycosylation showed a significant low field displacement of about 5–10 ppm. Unsubstituted C6 carbons (HO**C**H_2_ units) resonated from 60–63 ppm, and C6 carbons bound to another saccharide residue appeared around 65–70 ppm. The carbon signals of methyl groups were in the range of 15 to 30 ppm, and carbonyl carbons were in the range of 165 to 185 ppm [20].

The 1D ^1^H NMR spectrum showed the presence of several substructures indicative of polysaccharides according to chemical shift δ_H_ (cf. above). At δ_H_ 1.2–2.3 ppm, ^1^H, ^13^C DEPT HSQC NMR spectra showed multiple cross peaks indicative of methyl groups (δ_H_: 1.28, 1.30, 1.31, 1.38, 1.39, 1.45, 1.47, and 1.5 ppm) (Appendix A). A ^1^H, ^1^H TOCSY NMR spectrum provided a contiguous spin system connecting the following ^1^H NMR resonances (Appendix A) δ_H_ 1.28, 2.43, and 2.48 ppm, and δ_H_ 1.46, 1.52, 1.76, 1.82, 1.91, and 3.06 ppm, some of which were correlated to ^13^C signals of type CH_3_ on the HSQC spectrum and to very complex proton resonances in the ^1^H, ^1^H TOCSY NMR spectrum: δ_H_1: 1.28, 1.30, 1.31, 1.38, 1.39, 1.45, 1.47, and 1.5 ppm (Appendix A). In addition, 1.27/19 ppm (CH_3_) coupled to 4.28, 4.35, 4.45, and 4.48 ppm; 1.29 /19 ppm (CH_3_) coupled to 4.28 (^3^J(H6-H5): 8 Hz), 4.35, and 4.44 ppm; 1.30–1.31/22 ppm (CH_3_) (^3^J ~ 8 Hz) coupled to 2.47, 4.28 ppm, 1.32/22 ppm (low signal strength); (CH_3_) coupled (^3^J ~ 8 Hz) 2.48 ppm, 1.38/17.2 ppm coupled to 3.52, 3.76 ppm, 1.42/15.8 ppm (CH_3_) coupled to 4.68 ppm (^3^J ~ 8 Hz), 1.46/16.5 ppm (CH_3_) (^3^J ~ 8Hz) coupled to 4.38 ppm. The values in between 3.2 to 4.5 ppm correspond to the chemical movement of protons in a saccharide cycle. Resonance signals at 1.27, 1.29, 1.38, and 1.42 ppm indicate the distinctive chemical displacements of methyl 6-deoxysaccharides [21].

The positioning of cross peaks (δ_H/C_) of methyl from 6-deoxy-rhamnose in ^1^H, ^13^C DEPT HSQC NMR spectra is conspicuous and very sensitive to its environment. For example, methyl (C/H) resonates at 1.34–1.32/17.7–17.9 for one 4)-α-L-Rha*p*-(1→) [22] and 1.28/19.2 ppm [23] for another 4)-α-L-Rha*p*-(1→). δ_H/C_ indicated that the two methyl resonances at δ_H_ 1.27 and 1.29 connected with δ_C_ at 19 ppm could represent methyl in a →2-α-Rha*p*-1→ (1.28/19.2 ppm) [21]. The 1.38/17.2 ppm (δ_H/C_) de-shielded CH_3_ cross peak would represent binding of a α-rhamnose to a uronic acid [21,24]. The de-shielded proton signal and the shielded methyl carbon signal at 1.42/15.8 ppm (δ_H/C_, Appendix A) may indicate the presence of an adjacent nitrogen heteroatom. Signals resonating at 1.28, 1.30, 1.32, 1.41, 1.46, and 1.48 ppm could correspond to CH_3_ of amino acids such as, e.g., alanine, threonine, and valine.

A complex multiplet at 2.05 ppm and a broad massif at 1.94 ppm (CH_2_, 27.5 ppm and CH_2_, 28 ppm, respectively) indicated that these two signals are coupled (^3^J ~ 7 Hz), characteristic of a succinyl, substituting a saccharide via an ester bond [23]. A singlet at 2.10 ppm (CH_3_, 22.4 ppm) and an intense singlet at 2.17 ppm (CH_3_, 22.4) corresponded to acetate- or amide-type CH_3_. This assessment is corroborated by the presence of a 5.60 ppm (CH, 92.5 ppm) doublet (^3^J ~ 8 Hz) characteristic of a proton NMR resonance present on the carbon of an O-acyl glycosidic ring [25,26]. The highly dubbed chemical displacement excluded that this O-acetyl group could be carried by the C6 carbon of a hexose residue. In addition, the 8 Hz coupling constant reflected the axial–axial coupling of these protons, a 2.76 ppm Me (CH, 33 ppm) singlet, potentially H_2_ from a glucosamine. For example, a 3-O-Me on a galactose resonated at 3.46 ppm [27], and 3.45 and 3.42 ppm for a 2-OMe and 3-OMe on a 6-deoxyhexose [28]. Several signals correlated according to the 1H, 1H TOCSY NMR spectrum at 1.55 (CH_2_, 22 ppm), 1.75 (CH_2_, 26 ppm), 1.85 and 1.93 (CH_2_, 30.5 ppm), 3.06 (CH_2_, 39 ppm), and 4.37 (CH, 50 ppm) (Appendix A) indicated that a lysine was engaged in an amide bond, and similar properties were also observed in the LPS of *Proteus mirabilis* [29,30].

The ^1^H, ^13^C DEPT-HSQC cross peaks spectrum revealed directly connected carbon–proton pairs (^1^J_CH_) and differentiated CH/CH_3_ groups with positive signals (shown in red) from CH_2_ groups with negative signals (shown in blue) (Appendix A). Quaternary carbon, with no directly attached protons, was not visible in this spectrum [20]. The presence of cross peaks from δ_C_ 50–60 ppm (4.33/54, 4.52/55.5, 4.50/56, 4.36/58.5, 4.48/59 ppm, δ_H/C_) probably represented the presence of carbon bound to nitrogen. Several cross peaks from δ_H/C_ 80–85 ppm (4.18/81, 4.1/82.5, 4.08/82.5, 3.63/82, and 3.66/82 ppm, δ_H/C_) indicated the presence of furanose cyclic C [31]. The C1 of these furanoses resonated at 5.13/107 ppm (denoted l_2_), 5.15/107 ppm (denoted l_1_) and 5.098/107.5 (denoted l_3_) (Appendix A). All other signals in the range of cyclic CH chemical displacements corresponded to pyranoses.

In the area of the anomeric ^1^H and ^13^C resonances (4.506–5.278 ppm and 95.39–108.22 ppm), several types of anomeric carbons were identified from the most to the least unshielded displacement (Appendix A). The chemical shifts δ_H/C_ of respective HSQC cross peaks are summarized in Appendix A.

The joint analysis of several complementary NMR spectra led us to propose the following attributions. Resonances *δ**_H/C_* were unshielded resonances reflecting monosaccharide grafting or substitution. Resonance *δ**_H/C_* l_2_ (Appendix A): H1 5.13/107 ppm (J_H1,H2_ <2 Hz), H2 4.18/81 (J_H2,H3_ 16 Hz), H3 4.28/71 (J_H3,H4_ 12 Hz), H4 4.08/82.5 (J_H4,H5_ 10 Hz), H5 3.88, and 3.705/62.5 ppm represented the -2)-β-Ribofuranosyl-(1→ (H1-H5 correlation identified from TOCSY, HSQC NMR spectra). Resonances *δ**_H/C_* l_1_ (Appendix A): H1 5.15/107 (J_H1,H2_ 8 Hz), H2 3.84/75 (J_H2,H3_ 15 Hz), H3 3.56/76 (J_H3,H4_ 7.2 Hz), H4 4.10/82.5 (J_H4,H5_ 8 Hz), H5 3.58, and 3.64/62 represented the β-Ribofuranosyl-(1→ (identified from TOCSY, HSQC NMR spectra) [32]. Resonance *δ**_H/C_* l_3_ (Appendix A): 5.09/107.5, H2 4.19/81 (J_H1,H2_ 10 Hz), H3 4.27/71(J_H2,H3_ 8 Hz), H4 4.1/83 (J_H3,H4_ 10 Hz), H5 3.63, and 3.66 (J_H4,H5_ 12 Hz)/82 ppm represented -5)-β-Ribofuranosyl-(1→. The compilation of these NMR data gave a motif of type →5-(β-Rib*p*-2-β-Rib*p*)-1→2-β-Rib*p*-1→ which confirms the FT-ICR-MS data on the presence of tri-pentoses. Resonance *δ**_H/C_* j_1_ (Appendix A): the monosaccharide corresponding to H/C anomeric position j_1_ (Appendix A) presents the chemical displacements and coupling constants. The coupling constant ^3^J_H1,H2_ was lower than 5 Hz while the coupling constants ^3^J_H2,H3_, ^3^J_H3,H4_, and ^3^J_H4,H5_ were higher than 5 Hz, which evoked an α-Glcp. The chemical displacement of C1 (95 ppm) and the de-shielding of C2 (78 ppm) and C4 (76 ppm) led to the proposition of a -2,4)-α-Glc*p*-(1→ motif for proton j1 (Appendix A). Resonance *δ**_H/C_* j2 (Appendix A): proton j2 at 5.15/95 ppm correlated with H2 4.185/ (74 or 81 ppm) (^3^J < 5 Hz) and 4.29/71 ppm. For analogous reasons as those mentioned above, the proton j2 indicated a galacto-configuration. The chemical shift of j_2_ is potentially unshielded with the possibility of methoxy substitution in position 2. The coupling constants and the chemical displacement of the H1 and H4 protons suggest a →4)-α-Gal*p*-2-OMe-(1→. Resonance *δ**_H/C_* b_1_ (Appendix A): The H1 b_1_ (Appendix A) cross peak indicated a gluco- configuration, all the H of the ring showed TOCSY cross peaks with H1 and the coupling constants ^3^J were higher than 5 Hz. *δ**_H/C_* at position 1 (4.61/101 ppm) evoked a β-Glc*p*. δ_C_ of CH2 in position 6 was potentially unshielded (71 ppm), which suggested a sequence 6)-β-Glc*p*-(1→. The de-shielding of C3 and C4 carbon atoms (83 ppm) likely indicated a substitution of positions 3 and 4 by OAc. Glycoside b_1_ (Appendix A) could be a 6)-3,4-Di-OAc-β-Glc*p*-(1→, confirming the mass spectrometry hypothesis of a di-O-acetylated monosaccharide in 3,4 and bound in position 6. The proton h_1_ (Appendix A) could be a 3)-Rha*p*-(1→. Resonance e_1_ (Appendix A) was assigned, based on the H-indexed proton at δ_H_ 4.62 ppm, the correlation of H1 with H5 in the TOCSY NMR spectrum, the absence of a C6 resonance and the de-shielding of C4 and C5 carbons all together suggested an acid beta-glucuronic unit bound in 1 and 4 positions. This monosaccharide could be the unit linked to the amino acid lysine, previously identified in mass spectrometry. The NMR chemical shift of GlcA was similar to those of the pattern →4-β-D-Glc*p*A-6Lys-(1→ identified by Shashkov et al. in *Proteus mirabilis* LPS [33]. Based on these findings, we propose the following identification: -4)-β-D-Glc*p*A-6Lys-(1→. Overall NMR analysis confirmed the presence of OAc, methoxy substituents and the presence of succinate, of two deoxyhexoses (rhamnoses) and an amino acid-bound to uronic acid. This compound has the gross formula C_72_H_112_O_52_N_2_ and a partial putative structure of the *R. tataouinensis* TTB310 EPS has also been proposed (Figure 6).

### 2.7. Analysis of EPS Substituents by FTIR

FTIR analysis was performed to explore the substituted moieties along with monosaccharides. The standard infrared absorption peaks were observed in the range from 4000–400 cm^−1^ (Figure 7). 

A broad peak at 3342 cm^−1^ corresponded to the hydroxyl groups (-OH) of the monosaccharides and possibly to the ν(OH) mode of residual water in the sample. The peak at 2927 cm^−1^ corresponded to the ν(CH) stretching modes of methyl and methylene reported for polysaccharides [32,33]. The two intense and broad bands at 1025 cm^−1^ and 1066 cm^−1^ corresponded to C-O-C and C-O bonds, typical of monosaccharides and therefore strongly corroborated that the polymer is a polysaccharide [32,33]. The presence of lipids was not totally excluded, since the peak at 1751 cm^−1^ is characteristic of the ν(C=O) mode of ester bonds (C=OOR) [34]. However, there was no significant contribution of aliphatic groups, and the peak at 2927 cm^−1^ typical for the C-H bond is rather small. A better explanation for this band at 1751 cm^−1^ is the presence of O-acetylated sugar moieties. Bands at 1751–1735, 1248–1235, and 1377–1371 cm^−1^ have been convincingly assigned to ester groups -CH_2_-OCOCH_3_ resulting from the-acetylation of hydroxyl groups of polysaccharides [35,36]. In particular, bands at 1751, 1371, 1235, and 1047 cm^−1^ were assigned to the acetylation of glucose units in konjac glucomannan [36]. The bands observed at 1751, 1371, and 1236 cm^−1^ in the FTIR spectrum are thus in line with the acetylation of sugar moieties. A band may also contribute at 1047 cm^−1^, but probably superimposes with the dominating contributions from the sugar ring modes at 1066–1026 cm^−1^. The FTIR data corroborate the presence of OAc moieties in the EPS structure, as suggested by NMR and FT-ICR-MS analyses. The FTIR data are also in line with the presence of glucuronic acid, since β-D-glucuronic acid is characterized by intense bands at 1729 cm^−1^ (νC=O), 1089–1065 cm^−1^ (δCO + νCC), and 1024 cm^−1^ (τCO + νCO), which correspond to bands present in the FTIR spectrum of the sample [37]. The ν(C=O) mode of glucuronic acid as well as contributions from succinyl carboxylic groups could contribute to the infrared absorption observed at 1678–1720 cm^−1^. Infrared signatures from methoxy groups resulting from sugar methylation are expected at 1290–1270 and 1050–1010 cm^−1^ [38]. These bands are not expected to be intense. They cannot be assigned unambiguously in the FTIR spectrum, since contributions for the sugar moieties are also observed at these frequencies.

The typical band of peptide amide II mode ν(CN) + δ(N-H) was not present, in the spectrum, which indicated the absence of amide groups and peptides in the purified compound.

Although systematic studies have identified characteristic bands of some pentoses or (deoxy)hexoses in polysaccharides [39,40], superposition of contributions of the different sugars in the 1100–1000 cm^−1^ region makes it impossible to identify them in the sample. Glucose and galactose have specific contributions at 840 cm^−1^ [39]. Rhamnose disaccharide has an absorption maximum in this region at 1006 cm^−1^ [41]. Bands at 813–806 cm^−1^ have been associated with the presence of mannose in polysaccharides [39].

According to the literature, the absence of bands at ≈890 cm^−1^ and the bands at circa 840 cm^−1^ indicated the absence of β-glycosidic bonds and supported the presence of α-linked glycosidic units [42]. The strong band at 677 cm^−1^, however, could indicate the presence of a β(1-2)- bond linkage, as it was selectively observed in a (1-2)-linked-β-D-xylopyranoside in the crystalline (dry) state [43].

Overall, FTIR analysis of the EPS of *R. tataouinensis* TTB310 corroborated the presence of monosaccharides with O-acetyl and uronic acid type substitutions and suggested mainly the presence of α-glycosidic linkages.

## 3. Conclusions

In this study, optimization of culture conditions (composition of the medium, pH regulation) for the enrichment of the cyst biomass of *R. tataouinensis* TTB310 was successfully developed. The EPS, firmly attached to the bacterial surface, was productively extracted using TCA. According to elemental analysis and FTIR, the extracted EPS consists of saccharide units with the possible presence of amino acids. The monosaccharides involved in its structure, according to GC analyses, are ribose, glucose, galactose, and mannose. While glucose and galactose are the most common monomers among bacterial polysaccharides, ribose is rarely described as a component of repeated units of bacterial EPS. The FT-ICR-MS and NMR analyses also confirm the presence of ribose, which is the unique feature of the extracted EPS. Biochemical analysis confirms the EPS of *R. tataouinensis* TTB310 consisting of ten saccharide units (decasaccharide), which includes rare monosaccharides such as ribose and rhamnose, the amino acid lysine, and acetyl, succinyl, and methyl substitutions. The EPS of *R. tataouinensis* TTB310 constitutes the outer layer of the cyst. It is therefore responsible for cell defense mechanisms in general and in particular, in the context of the semi-arid conditions in which this bacterial strain was isolated, this EPS allows it to adapt to desiccation. Its structure is unique and completely different from that of other bacterial cysts such as *Azotobacter*, which is an alginate [44], and also different from the well-known succinoglycan, xanthan, gellan, pullulan, hyaluronan, and cellulose produced by soil bacterial species [45]. Some details remain to be confirmed with 1D ^13^C NMR, NOESY, and HMBC NMR spectra that would identify the sequence and assembly of the identified units in the spectrum. The purified polymer was insoluble in water or saline even after multiple treatments. However, it can be easily solubilized in DMSO. This insolubility in water is probably due to the presence of acetyl groups in the EPS structure. The obtained results encourage further studies on the deacetylation of the EPS of *R. tataouinensis* TTB310 which would possibly initiate structural changes allowing for its solubilization, thereby enhancing its rheological properties for industrial applications.

## 4. Materials and Methods

### 4.1. Bacterial Strain and Culture Conditions

*Ramlibacter tataouinensis* TTB310^T^ (=DSM 14655^T^ =ATCC BAA-407^T^ =LMG 21543^T^), a β-proteobacterium isolated from weathered particles of “Tataouine meteorite” embedded by sand particles was used in this study [8]. This strain was grown on Tryptic Soy Agar (TSA 1/10) medium (10 times diluted Bacto™ Tryptic Soy Broth, agar, 15 g/L) at 30 °C with optimal pH about 7.5. The strain TTB310 was maintained in TSA 1/10 plates at 4 °C and used for further EPS production.

### 4.2. Optimization of Cyst Formation by R. tataouinensis TTB310

The cyst formation was optimized by batch and fed-batch conditions. Initially, carbon sources were tested under batch conditions (500 mL Erlenmeyer flasks containing 100 mL of culture medium) using TSB 1/10 medium in the presence of acetate, succinate, β-hydroxybutyrate, and lactate (10, 20, 40 mM) as a sole carbon source, and the cultures were incubated at 30 °C for 3 days under 150 rpm agitation. The bacterial growth was estimated by counting the colony-forming units (cfu/mL), and colony morphology was also studied. Further optimization was performed in 30 L-bioreactor containing 9 L of culture medium in fed-batch condition. In order to control the pH increase and to limit the osmotic stress, lactic acid (10%) was used with a set point value of pH 7.4. The optimized medium was composed of TSB 1/10 + lactate (10 mM), tryptone (3.0 g/L), yeast extract (1.3 g/L), K_2_HPO_4_ (0.25 g/L), and NaCl (0.5 g/L). After 4 days of cultivation at 30 °C, the medium was further supplemented with 1.0 g/L of tryptone and 0.3 g/L of yeast extract. The sterile air supply set point was 0.2 vvm, and an agitation rate of about 1.98 m/s was maintained during the overall period of fermentation.

### 4.3. Production, Extraction and Purification of EPS from Cysts

The initial seed culture (inoculums) was prepared by inoculating 5 mL of sterile TSB 1/10 using a single colony *of R. tataouinensis* TTB310, and it was grown for 3 days under shaking with a speed of 150 rpm at 30 °C. About 1% (v/v) of inoculum was seeded into a 2 L Erlenmeyer shake flask containing 500 mL of sterile EPS production medium, which is comprised of TSB 1/10, lactate (10 mM), tryptone (3.0 g/L), yeast extract (1.3 g/L), K_2_HPO_4_ (0.25 g/L), and NaCl (0.5 g/L). The pH of the medium was fixed to 7 ± 0.2 and the strain TTB310 was cultured under growth conditions at about 30 °C and at a shaking speed of 150 rpm for 4 days. About 1 mL of fermented culture was intermittently collected to check the cyst formation and EPS synthesis during the fermentation process.

After 4 days of cultivation, the spent culture was centrifuged at 6000× *g* for 15 min at room temperature and the resulting bacterial pellet was liquefied in 25 mL of modified phosphate-buffered saline (PBS: KCl 5.0 g/L, Na_2_HPO_4_ 1.44 g/L, KH_2_PO_4_ 0.24 g/L) supplemented with 4% trichloroacetic acid (TCA). The mixture was kept under magnetic stirring for 12 h at room temperature. The solution was then centrifuged at 6000× *g* for 15 min at room temperature. The supernatant was collected and filtered using a 0.45 µm Sartorius Minisart high flow syringe filter. The filtrate was neutralized (pH 7.0) with 1 M KOH, and 10 volume of pre-chilled absolute ethanol was added into neutralized filtrate, and this mixture was incubated at 4 °C for 12 h. A white polymer precipitate was then observed, and the whole blend was centrifuged at 6000× *g* for 20 min at 4 °C. To know the efficiency of this modified extraction, cells were also stained by calcofluor white and observed under optical microscopy. The EPS polymer was then washed with different concentrations of ethanol (70%, 80%, 90%, and 100%) and the polymer was further purified by overnight dialysis using MWCO 1 kDa Spectra/Por 7 dialysis section. Dialyzed EPS (0.1%) samples were prepared with 0.1 M NaCl and ultrapure water, and the pH was adjusted to 7.0 or 10.7. All the samples were shaken at 40 °C, and the temperature was raised to 60 °C after 30 min or 24 h. Further, the EPS samples were subjected to several sonications at 40 °C to 90 °C for 30 s and 15 min. The concentration of EPS was made about 0.1% in dimethyl sulfoxide (DMSO) and stirred and subsequently heated at 60 °C in order to solubilize the EPS. The final concentration was adjusted to 4%.

### 4.4. Chemical and Structural Characterizations of the EPS

#### 4.4.1. Chemical Analysis

Elemental analysis for the determination of C, H, N, and O in the extracted EPS was performed according to the modified protocol with an organic element analyzer (Thermo Scientific FLASH ™ 2000 CHNS/O, Thermo Fisher Scientific, Massachusetts, USA). The determination of the amount of C, H, and N was carried out by total pyrolysis of the analytical sample at 1800 °C in the presence of oxygen. The oxygen content was detected by total pyrolysis of the EPS sample at 1080 °C under a stream of nitrogen. The total chlorine was quantified by using high-pressure ion chromatography (HPIC, Waters). Total monosaccharide content was assayed by phenol-sulfuric acid methods with glucose as standard [46]. Uronic acids were quantified using the colorimetric method of meta-hydroxydiphenyl [47]. RNA detection was performed using the Nanovue spectrophotometer (GE Healthcare) and also verified with the “Quant-it™ RiboGreen^®^” kit (Invitrogen, Waltham/ Massachusetts, USA).

#### 4.4.2. Fourier-Transform Infrared Spectroscopy (FTIR)

FTIR Spectroscopy (IFS 66 SX, Bruker Corporation, Ettlingen, Germany) was used to analyze the main functional groups in the purified, freeze-dried EPS of *R. tataouinensis* TTB310. FTIR spectra were recorded in the in 4000–400 cm^−1^ region with a resolution of 4 cm^−1^ using a SensIR technologies ATR device equipped with a 9-bounce diamond microprism and ZnSe optics.

#### 4.4.3. High-Performance Liquid Chromatography (HPLC) Analysis

Neutral lipid moieties that are associated with the EPS were detected and fractionated by HPLC system (UltiMate 3000 RS, Dionex, France) coupled with a Sedex 85 evaporative light scattering detector (Sedere S.A., Alfortville, France) with eluent A (chloroform) and eluent B (CH_3_OH/CHCl_3_/NH_4_OH). The lipid classes were detected and quantified by an evaporation light scattering detector (ELSD, SEDEX Model 85 LT-ELSDTM, Sedere, Alfortville, France). Comparing their retention time with those of pure standards was performed in order to identify the lipid classes.

#### 4.4.4. Gas Chromatography (GC) Analysis of EPS

Gas chromatography analysis was performed after methanolysis. The EPS sample (3.2 mg) and 50 μg of myo-inositol (internal reference) were added into 500 μL of a mixture of methanol/hydrochloric acid and incubated in a dry bath for 4 h at 110 °C. After cooling to room temperature, the methanolysate was neutralized with silver carbonate, and the samples were centrifuged at 3000 rpm for 15 min at 4 °C. The supernatant was evaporated under nitrogen gas. The compounds were then dissolved in 100 μL of pyridine and incubated overnight at room temperature with 100 μL Cylon (BSTFA: TMCS. 99:1. Supelco). After gentle evaporation of the excess of reagents under nitrogen gas, the trimethyl-silylated methyl glycosides were taken up in 700 μL dichloromethane, then injected into gas chromatography (GC-6850 AGILENT System, in-column injection, FID detector: flame ionization) with hydrogen as carrier gas. The temperature profile was as follows: 120 °C maintained for 1 min, then a gradient of 1.5 °C/min up to 180 °C, followed by a gradient of 2 °C/min up to 200 °C. Each monosaccharide was identified by comparison with the retention times of the internal standards treated under the same conditions. A response coefficient was calculated for each monosaccharide relative to the internal standard to define their proportion within the polysaccharide.

#### 4.4.5. Fourier Transform Ion Cyclotron Resonance Mass Spectrometry (FT-ICR-MS)

Sample duplicates were dissolved in 2 mL H_2_SO_4_ 5% (Merck, Germany) and sonicated for 3 min at room temperature. Then the samples were passed through SPE-C18 cartridges for FT-ICR-MS measurements. Negative electrospray (-)ESI FT-ICR MS were acquired using a mass spectrometer (Bruker Solarix 12T model, Bruker Daltonics, Bremen, Germany). The samples diluted in methanol were injected into the electrospray source using a pump with a flow rate of 120 μL/h, a nebulizer gas pressure of 138 kPa and a drying gas pressure of 103 kPa. Heating at 200 °C was maintained to ensure rapid desolvation of the ionized droplets. Each acquisition was taken in the time domain of 4 MW, with 500 scans being accumulated for a single mass spectrum. All spectra were internally calibrated using an appropriate reference mass list in order to obtain a mass accuracy of less than 0.2 ppm. Data processing was conducted using Compass Data Analysis 4.1 (Bruker, Bremen, Germany) and formula assignments were processed by in-house software. The generated formulae were validated by setting sensible chemical constraints [O/C ratio ≤ 1, H/C ratio ≤ 2n + 2 (C_n_H_2n+2_)], element counts (C ≤ 100, H ≤ 200, O ≤ 80, N = 0, S = 0) and mass accuracy windows (set at 500 ppb). Final formulae were generated and categorized into CHO-containing molecular compositions which were used for the evaluation.

#### 4.4.6. Nuclear Magnetic Resonance (NMR)

The EPS sample was subjected to liquid-state NMR spectroscopy. About 4 mg of dry sample was diluted in 750 μL of D_2_O (99.95%) under ultrasonic bath with external heating about ~ 45 °C. After centrifugation, the supernatant was subjected to 800 MHz 1H NMR spectroscopy at 310 K. Homo-nuclear J-resolved spectroscopy (JRES), hetero-nuclear single-quantum correlation (HSQC), correlation spectroscopy (COSY), and total correlated spectroscopy (TOCSY) were acquired for the detailed analysis of the structure of the EPS.

A Bruker Avance III spectrometer and TopSpin 3.5/PL7 software were used to acquire nuclear magnetic resonance (NMR) spectra of aqueous extracts of a white, semitransparent, and potentially hygroscopic (by appearance) pellet of polymer. About 4 mg of white, semitransparent, and potentially hygroscopic (by appearance) pellet were subjected with 750 µL D_2_O (99.95%, Aldrich) to ultrasonic bath, with external warming to ~45 °C. After 5 h of treatment, the sample was centrifuged, and the supernatant was put into a 3.0 mm MATCH tube and sealed.

A cryogenic inverse geometry 5 mm z-gradient ^1^H/^13^C/^15^N/^31^P QCI probe (B0 = 18.8 T) was used for 1D ^1^H NMR and proton-detected 2D NMR spectra. Transmitter pulses were at ~10 µs for ^1^H and ^13^C. The one-bond coupling constant ^1^J(CH) used in 2D ^1^H,^13^C DEPT-HSQC spectra (hsqcedetgpsisp2.2) was set to 145 Hz. Other conditions: ^13^C 90 deg decoupling pulse, GARP (70 µs), 50 kHz WURST 180-degree 13C inversion pulse (Wideband, Uniform, Rate, and Smooth Truncation, 1.2 ms), F2 (^1^H): spectral width of 11160.7 Hz (13.95 ppm), 1.25 s relaxation delay, F1 (^13^C): SW = 36052 Hz (180 ppm). HSQC-derived NMR spectra were computed to an 8192 × 1024 matrix. Gradient (1 ms length, 450 µs recovery) and sensitivity enhanced sequences were used for all 2D NMR spectra. The absolute value COSY and phase-sensitive echo-antiecho TOCSY spectra (cosygpmfppqf, dipsi2etgpsi) used a spectral width of 9615.4 Hz and were computed to a 16384 × 2048 matrix; other NMR acquisition conditions are given in Appendix A.

## Figures and Tables

**Figure 1 molecules-27-07172-f001:**
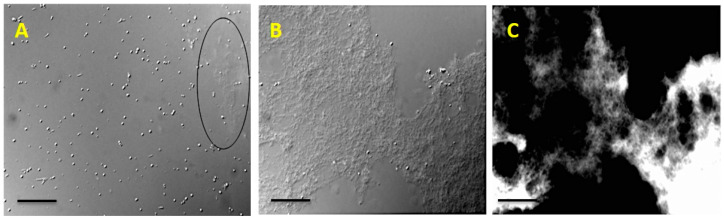
Detachment of EPS from the surface of *R. tataouinensis* TTB310 cells after suspending in modified PBS and 4% TCA treatment. (**A**) Optical microscopy image in which calcofluor white staining reveals a filamentous fluorescent part (cut-off area). (**B**) Optical microscopy image of a large fragment of EPS taken off. (**C**) The fluorescence image after staining with calcofluor white corresponding to image B, which shows the fluorescence of the peeled polymer. Scale bar: 20 μm.

**Figure 2 molecules-27-07172-f002:**
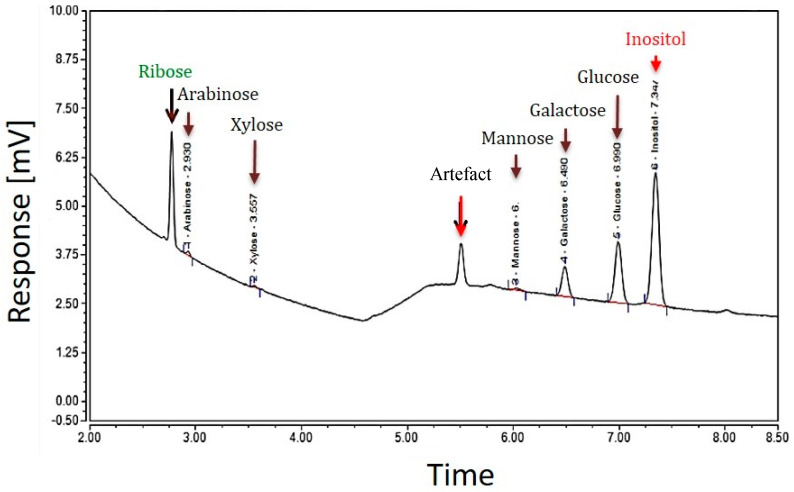
GC profile of the EPS of *R. tataouinensis* TTB310 which were pre-treated by acid hydrolysis (13 M H_2_SO_4_): The major peaks detected correspond to the monosaccharides: ribose, glucose, and galactose, and the minor peaks are those of xylose, arabinose, mannose. The internal standard was inositol.

**Figure 3 molecules-27-07172-f003:**
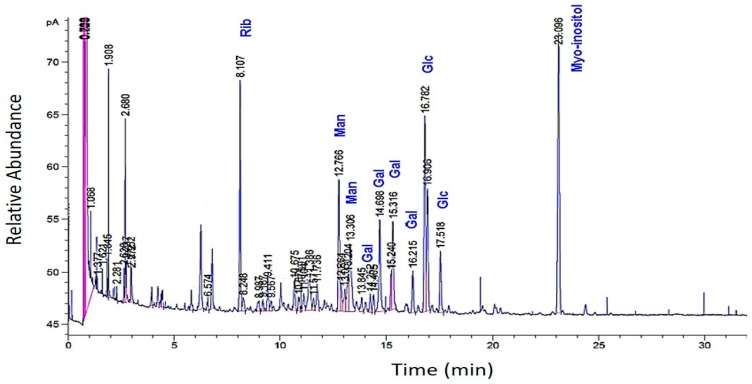
GC profiles of the EPS of *R. tataouinensis* hydrolyzed by methanolysis (MeOH/HCl 3 N). The relative abundance of the monosaccharides: ribose, glucose, galactose, and mannose referred to the internal standard myo-inositol.

**Figure 4 molecules-27-07172-f004:**
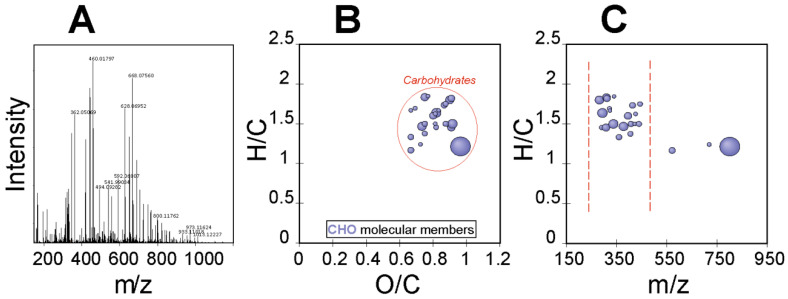
Detected carbohydrates in the *R. tataouinensis* EPS sample as detected by FT-ICR-MS: (**A**) Full mass spectrum, (**B**) van Krevelen diagram, and (**C**) mass-edited H/C ratios plot.

**Figure 5 molecules-27-07172-f005:**
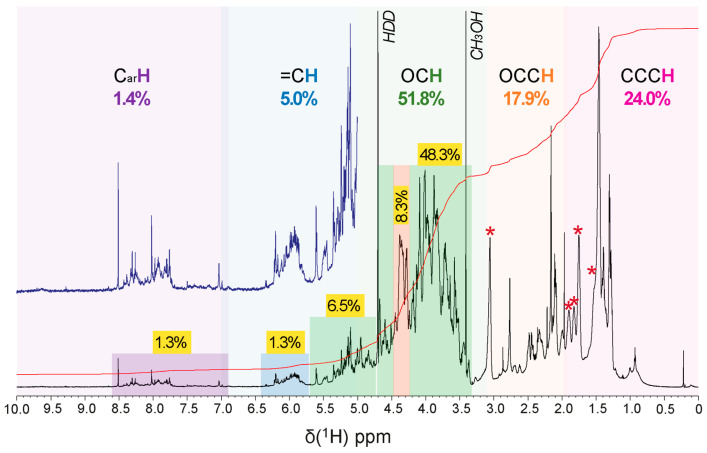
^1^H NMR spectrum (800 MHz, D_2_O) of *R. tataouinensis* EPS reveals a considerable structural complexity, with the key substructures (C_ar_**H**, purple, =C**H** and O_2_C**H**, blue, OC**H**, green, NC**H** and OCC**H**, orange, and CCC**H**, red) and relative abundance (% of 1H NMR section integral) indicated. A stronger green shade denotes carbohydrate-related OC**H** and O_2_C**H** units, a red shade probably indicates peptide CONHCα**H** units, a blue shade denotes olefinic =C**H** units, and a purple shade denotes aromatic C_ar_**H** units; asterisk denotes an extended aliphatic spin system, cf. Appendix A.

**Figure 6 molecules-27-07172-f006:**
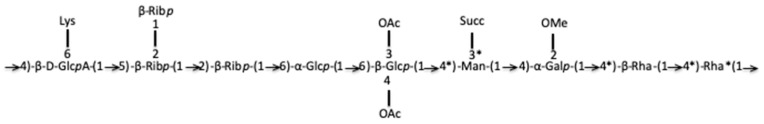
Partial putative structure of the purified EPS obtained from the cysts of *R. tataouinensis* TTB310. The C*s are those for which it is necessary to check that they are well engaged in the link. Lys: Lysine; OAc: O-acetyl group; Succ: succinate; OMe: O-methyl group.

**Figure 7 molecules-27-07172-f007:**
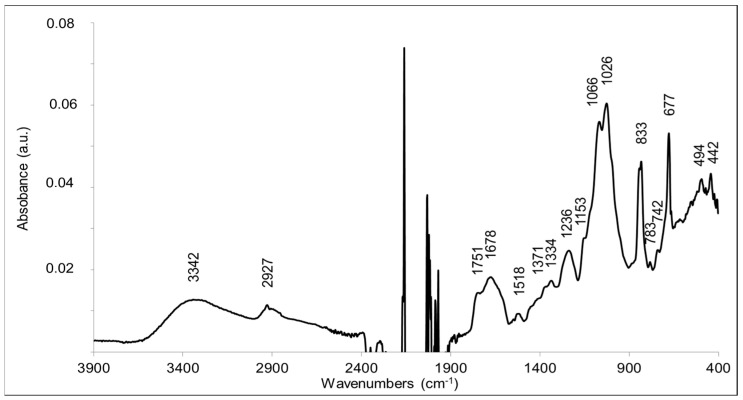
FTIR spectrum of the extracted EPS from *R. tataouinensis* TTB310 cysts. The spectrum was recorded in the 3900–400 cm^−1^ range with a 4 cm^−1^ resolution using an ATR device (SensIR technologies, CT) equipped with a 9-bounce diamond microprism and ZnSe optics. The 2400–1900 cm^−1^ region cannot be exploited due to the absorption of CO_2_ and the strong absorption of the diamond material.

## Data Availability

Not applicable.

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
