# Peer review of "Production and Characterization of a Novel Exopolysaccharide from *Ramlibacter tataouinensis"

_molecules, 2022, doi:10.3390/molecules27217172_

Round 1
Reviewer 1 Report
The current study is set to determine the molecular characterization of a novel exopolysaccharide (EPS) produced by Ramlibacter tataouinensis by gas chromatography, ultrahigh-resolution mass spectrometry, nuclear magnetic resonance and Fourier-transform infrared. This is original research performed on a novel exopolysaccharide from a dividing form of cysts using advanced technologies. This study’s findings would draw the attention of scientific and industrial communities because of its novelty and potential industrial applications. On the other hand, there are some remarks given below that should be taken into consideration to make some points clear:
1. EPS molecules are involved in cell defense mechanisms to cope with osmotic, saline, low/high-temperature stress, desiccation, antibiotics, and antibodies. Based on the finding of this study, it would be better to state the biochemical characteristics of this novel EPS and its relation to the resistance to extreme environmental conditions in conclusion.
2. As stated in the results and discussion (lines: 112-114), previous studies have shown that R. tataouinensis assimilate acetate, lactate, and β-hydroxybutyrate as carbon and energy sources, but such stimulation was not observed in this study. This requires an explanation based on the current findings of this study.
3. I cannot see any substantial difference between the pH (8.7) at the end of the first experiment (line 129) and stopping the lactic acid supply beyond the 7th day (pH 8.4). It seems there are some flaws in this respect. Please check the explanations made on the findings of the addition of lactic acids (Lines 128-133).
4. HPLC analysis showed the presence of lipids in the total EPS samples. It has been stated that “This might arise from the extraction process since the EPS of strain TTB310 is tightly bound to outer membranes of the bacterial cells” It would be better to cite a reference/s showing similar results.
5. The finding of this study has been given in detail. However, the biochemical characteristics of this novel EPS were not compared with those previously characterized by other bacterial strains.
6. Species name in the title and abstract must be written in italics.
7. EPS must be written fully in the abstract.
Author Response
Answers (in blue) to the Comments and Suggestions for Authors
Reviewer 1
The current study is set to determine the molecular characterization of a novel exopolysaccharide (EPS) produced by Ramlibacter tataouinensis by gas chromatography, ultrahigh-resolution mass spectrometry, nuclear magnetic resonance and Fourier-transform infrared. This is original research performed on a novel exopolysaccharide from a dividing form of cysts using advanced technologies. This study’s findings would draw the attention of scientific and industrial communities because of its novelty and potential industrial applications. On the other hand, there are some remarks given below that should be taken into consideration to make some points clear:
- EPS molecules are involved in cell defense mechanisms to cope with osmotic, saline, low/high-temperature stress, desiccation, antibiotics, and antibodies. Based on the finding of this study, it would be better to state the biochemical characteristics of this novel EPS and its relation to the resistance to extreme environmental conditions in conclusion.
Thank you for this comment. We have modified the conclusion accordingly (Lines 551-555, in the corrected version):
“The EPS of R. tataouinensis TTB310 constitutes the outer layer of the cyst. It is therefore responsible for cell defense mechanisms in general and in particular, in the context of the semi-arid conditions in which this bacterial strain was isolated, this EPS allows it to adapt to desiccation. Its structure is original and completely different from that of other bacterial cysts such as Azotobacter which is an alginate.”
- As stated in the results and discussion (lines: 112-114), previous studies have shown that R. tataouinensis assimilate acetate, lactate, and β-hydroxybutyrate as carbon and energy sources, but such stimulation was not observed in this study. This requires an explanation based on the current findings of this study.
There was a weak increase of growth in presence of acetate, succinate or β-hydroxybutyrate but much smaller than that observed with lactate, probably because lactate regulates also pH medium and not acetate, succinate and β-hydroxybutyrate. We have modified this sentence (Lines 121-125, in the corrected version):
“Although previous studies where R. tataouinensis was shown to assimilate acetate, lactate, and β-hydroxybutyrate as carbon and energy sources [8,11], we could not observe such important stimulation of growth in presence of acetate, hydroxybutyrate or succinate, probably because only lactate can regulate pH medium (see # 2.2).”
- I cannot see any substantial difference between the pH (8.7) at the end of the first experiment (line 129) and stopping the lactic acid supply beyond the 7thday (pH 8.4). It seems there are some flaws in this respect. Please check the explanations made on the findings of the addition of lactic acids (Lines 128-133).
You are right. At the end of the experiment, when the lactic acid supply was stopped, the increase of pH medium was the same. In Figure S3B, we have evidenced that in presence of a continuous lactate supply, the pH medium was well regulated until the supply of lactic acid was stopped (about 160 h).
- HPLC analysis showed the presence of lipids in the total EPS samples. It has been stated that “This might arise from the extraction process since the EPS of strain TTB310 is tightly bound to outer membranes of the bacterial cells” It would be better to cite a reference/s showing similar results.
Very few EPS of bacterial cyst have been characterized apart from Azotobacter (alginate). The presence of EPS-associated lipids has not been described to date.
- The finding of this study has been given in detail. However, the biochemical characteristics of this novel EPS were not compared with those previously characterized by other bacterial strains.
Thank you for this remark. We have modified the conclusion accordingly (Lines 546-552, in the corrected version):
“Its structure is original and completely different from that of other bacterial cysts such as Azotobacter which is an alginate [45], and also different of the well-known succinoglycan, xanthan, gellan, pullulan, hyaluronan and cellulose produced by soil bacterial species [46].”
- Species name in the title and abstract must be written in italics.
Modification done.
- EPS must be written fully in the abstract.
Modifications done
Reviewer 2 Report
Dear Authors,
The manuscript present a lot of data – enough to description of EPS structure, however it should be rewrite in some section for be more clear and transparent for readers.
1. Title: in manuscript no information about EPS biosynthesis is included.
2. Abstract: please rewrite the first sentence.
3. What was a yield of purified EPS?
4. How The Authors have checked the purity of EPS?
5. EPS is “cell-bound” (line 22) or loosely associated to the bacterial cell (line 45)?
6. Line 53: please remove “ polymers” – EPS is a polymer
7. Lines 92, 117: cysts
8. Line 120: pH 7.4 not “about”
9. Line 133: pH 8.4
10. Line 156: “glycoprotein bonds”????
11. What is a proof for tightly EPS bound to outer membrane? The method for EPS extraction which has been used, should gave a result in purified EPS. In case the Authors observe a lipids in EPS...it means the EPS sample is not purity. Am I right?
12. After EPS extraction: all experiments (GC, NMR, FT-ICR-MS) have been done on the same EPS sample?
13. Line 191: “H2SO4”
14. Line 195: Please rewrite the last sentence
15. Line 200: please remove: ...and some peaks....internal standards”
16. Line 213: “several thousand mass peaks”???
17. m/z
18. Masses with two places after decimals
19. Line 259: please remove “polymer”, by NMR
20. Section 2.6. The all first paragraph is not necessary. Here, the most important things is presentation of results.
21. Line 279: “several substructures” what does mean? – it is a heterogeneity of the sample. So the question is – EPS sample is purified or not? In case EPS extractions, or even other polysaccharides/oligosaccharides the heterogeneity is one of the problem by the fact biosynthesis.
22. Please rewrite this section very carefully considering only observed facts. Compare the NMR spectra. Do Authors have got any NOESY/HMBC spectra? The complete structure should be confirmed by readable NMR data. Please prepare the NMR data more readable (Figures S6, S8, S9). I think all important information the Authors can read from NMR spectra.
23. What does mean asterisks?(figure 6)
24. Figure 6: please improve the figure with including letter markings of each residue. Additional substitutions by OMe, OAc, Lys, succinate please mark different than glycosidic bound between sugar residues.
25. Line 367: ...b1”??
26. Line 581: “neutral lipids....associated in the EPS... were ...fractionated” How many fractions were received? What were a differences between them? What exactly fraction was analyzed...the pure EPS?
27. Line 590: “about 1.8 and 3.2 mg” why about? And why two masses?
28. Line 594: what was a centrifugation temperature?
29. Line 622: remove “)”
30. Line 626: NMR spectroscopy
31. Lines 628, 638: D2O, please remove “2H”
Author Response
Answers to the Comments and Suggestions for Authors
Reviewer 2
The manuscript present a lot of data – enough to description of EPS structure, however it should be rewrite in some section for be more clear and transparent for readers.
- Title: in manuscript no information about EPS biosynthesis is included.
We have modified the title to take into account this comment: “Production and characterization of a novel exopolysaccharide produced by Ramlibacter tataouinensis”
- Abstract: please rewrite the first sentence.
We have modified the sentence accordingly: “The current study deals about the desiccation resistant Ramlibacter tataouinensis TTB310T as a model organism for the production of novel exopolysaccharides and their structural features.”
- What was a yield of purified EPS?
We have not calculated the yield of the EPS production after purification.
- How the Authors have checked the purity of EPS
The extraction and the purification of the EPS has been performed in independent triplicates, leading to the same characteristics of the samples, showing that a reproducible level of purity was reached.
- EPS is “cell-bound” (line 22) or loosely associated to the bacterial cell (line 45)?
The EPS of Ramlibacter tataouinensis TTB310T is the outer layer of the cyst, probably partly "cell bound". For most bacterial EPS (not constituting a cyst outer layer), it is generally accepted that this EPS is weakly associated with the bacterial cell.
- Line 53: please remove “ polymers” – EPS is a polymer
“polymers” was removed in Line 61, in the corrected version
- Lines 92, 117: cysts
Correction done in Lines 107 & 146, in the corrected version: “…also exhibited both cysts and rods”
- Line 120: pH 7.4 not “about”
We have deleted “about” in Line 129, in the corrected version
- Line 133: pH 8.4
We have modified “pH 8.4” instead of “pH (8.4) in Line 143, in the corrected version
- Line 156: “glycoprotein bonds”????`
Among the different protocols tested, the one using trichloroacetic acid was the most efficient to extract the EPS. However, we agree with reviewer 2 that there is no evidence that TCA cleaves the glycoprotein bonds. The sentence has been modified accordingly:
Lines 170-171, in the corrected version: “The modified extraction method using PBS with TCA has significantly increased the yield of EPS, since TCA particularly denatures and precipitates the proteins.”
- What is a proof for tightly EPS bound to outer membrane? The method for EPS extraction which has been used, should gave a result in purified EPS. In case the Authors observe a lipids in EPS...it means the EPS sample is not purity. Am I right?
Observations of electron micrographs of ultrathin sections from cysts have revealed the presence of a thick exopolysaccharide layer closely associated with the outer membrane (Heulin et al., 2003). Classically, during the purification of EPS, adding cold ethanol to the supernatant of the cell pellets allows to precipitate the exopolysaccharides away from the lipids. However, most of the studies do not quantify the lipid content in EPS after purification, which we did, to search for explanations of the very low solubility of the EPS in water. The extraction and the purification of the EPS has been performed in independent triplicates, leading to the same characteristics of the samples, showing that a reproducible level of purity was reached.
- After EPS extraction: all experiments (GC, NMR, FT-ICR-MS) have been done on the same EPS sample?
Yes
- Line 191: “H2SO4”
Line 205, in the corrected version: correction done
- Line 195: Please rewrite the last sentence
The sentence has been modified.
Lines 226-228, in the corrected version: “In our hands, methanolysis was found to be a less efficient hydrolysis method for this EPS compared to sulfuric acid hydrolysis. “
- Line 200: please remove: ...and some peaks....internal standards”
The sentence has been modified accordingly:
Lines 212-213, in the corrected version: “The GC analysis of methanolysed EPS revealed the presence of ribose, glucose, galactose and mannose. Interestingly, certain monosaccharides (e.g. arabinose, xylose) were absent (Fig. 3).
- Line 213: “several thousand mass peaks”???
The sentence has been modified:
Lines 230-231, in the corrected version: “The ultrahigh-resolution mass spectrometry (FT-ICR-MS) for the EPS of R. tataouinensis TTB310 between 100 and 1500 m/z is showed in Fig. 4A”.
- m/z
Lines 230 and everywhere else: correction done
- Masses with two places after decimals
FT-ICR-MS allows to determine m/z with a with a precision of 5 decimal places.
The measurement of fragment ions at resolving power and high mass accuracy allowed for the in-depth characterization of the analyzed synthetic compounds (Nicolardii et al., 2021).
Nicolardi, S., Joseph, A. A., Zhu, Q., Shen, Z., Pardo-Vargas, A., Chiodo, F., ... & Wuhrer, M. (2021). Analysis of synthetic monodisperse polysaccharides by wide mass range ultrahigh-resolution MALDI mass spectrometry. Analytical chemistry, 93(10), 4666-4675.
Line 259: please remove “polymer”, by NMR
Line 307, in the corrected version: correction done
- Section 2.6. The all first paragraph is not necessary. Here, the most important things is presentation of results.
We disagree with Reviewer 2 about the uselessness of the first paragraph of section 2. This paragraph identifies the signals corresponding to carbohydrates, signals on which we rely for the interpretation of the EPS structure. The EPS was purified. However, the complexity of R. tataouinensis cysts, which have the peculiarity of dividing, the strong association of the EPS with the outer membrane, and the low solubility of the EPS strongly limit the possibility of further purification of the sample. Despite this difficulty, saccharide identification by GC after hydrolysis and lipid quantification remained consistent over three purified samples from three independent extractions.
From the ability to identify peaks corresponding to carbohydrate structures by FT-ICR-MS based on the C/H ratio, and from the neutral mass loss patterns and potential structures assigned to the m/z list, it is possible to reconstruct the structure of potential structures. NMR completes this interpretation by providing further information on the sequence of these structures.
- Line 279: “several substructures” what does mean? – it is a heterogeneity of the sample. So the question is – EPS sample is purified or not? In case EPS extractions, or even other polysaccharides/oligosaccharides the heterogeneity is one of the problem by the fact biosynthesis.
We agree with Reviewer 2 that the biosynthesis of polysaccharides can lead to heterogenous structures. As previously mentioned, the EPS was purified as thoroughly as possible, given its low solubility, and the purification was achieved in a reproducible way.
- Please rewrite this section very carefully considering only observed facts. Compare the NMR spectra. Do Authors have got any NOESY/HMBC spectra? The complete structure should be confirmed by readable NMR data. Please prepare the NMR data more readable (Figures S6, S8, S9). I think all important information the Authors can read from NMR spectra.
The spectra were interpreted and described accurately, with chemical shifts, coupling constants, and interpretation based on the confrontation of all the different spectra performed. The C and H chemical shifts of the molecule are listed in Table S7.
We did not perform NOESY/HMBC spectra, however we got several analysis based on:
- a 2D JRES (Homonuclear J-resolved Spectroscopy) spectrum, which is a J-resolved spectrum, allowing to obtain the proton chemical shift on one axis and the proton-proton coupling on the other
- a 2D HSQC (Heteronuclear Single-Quantum Correlation) spectrum which analyzes the correlation of the chemical shifts of 1H and 13C by the direct 1J couplings, and the hybridization of carbons (CH3, CH2 or CH). This spectrum is very useful to quickly identify structural elements.
- Different sequences 2D-1H1H, COSY (COrrelation SpectroscopY) and TOCSY (TOtal Correlated SpectroscopY) which allow to individualize the coupled spin systems and to measure the 3JHH coupling constants between protons. The order of magnitude of this coupling constant gives information on the stereochemistry of the protons and allows to determine the nature of the configuration, gluco-, manno- or galacto-.
The final structure proposed for the EPS relies on the confrontation of FT-IR-MS, RMN, IR spectra and all the results from the analytical analysis of the EPS (GC and HPLC). Even with the overall results, we remain cautious in proposing a putative structure.
- What does mean asterisks?(figure 6)
Some carbohydrates and their linkages are proposed on the basis of NMR but are not confirmed by all the other analyses. As a precautionary measure, these carbohydrates or their linkage are identified by asterisks. The legend of Fig. 6 has been completed accordingly: “The C*s are those for which it is necessary to check that they are well engaged in the link”
- Figure 6: please improve the figure with including letter markings of each residue. Additional substitutions by OMe, OAc, Lys, succinate please mark different than glycosidic bound between sugar residues.
The Figure 6 was improved by changing the symbol of the OMe, OAc, Lys, succinate links in the scheme, and adding the name of abbreviations.
- Line 367: ...b1”??
Lines 415-416, in the corrected version. You are right, there was a mistake: b1 instead of h1. The sentence has been modified accordingly.
- Line 581: “neutral lipids....associated in the EPS... were ...fractionated” How many fractions were received? What were a differences between them? What exactly fraction was analyzed...the pure EPS?
The purified EPS was analyzed. We analyzed the lipid fraction as a possible explanation to the low solubility of the EPS. Concerning the nature of these lipids, the HPLC analysis detected the presence of neutral lipids, notably triglycerides, monoglycerides and free fatty acids. About 1.5% of lipids on average were present in the samples. This indicates a low amount of fatty acids. The HPLC profile showed the major presence of lipids compatible with a partial extraction of the external membrane of the bacterial cells during the extraction of the EPS. According to this lipid quantification, the insolubility of the EPS is rather due to its structure, likely the presence of β-bonds.
- Line 590: “about 1.8 and 3.2 mg” why about? And why two masses?
Line 658, in the corrected version: corrected by “The EPS sample (3.2 mg)…”
- Line 594: what was a centrifugation temperature?
Line 662, in the corrected version. The centrifugation temperature was 4°C: correction done
- Line 622: remove “)”
Correction done in Line 690, in the corrected version
- Line 626: NMR spectroscopy
Correction done in Line 694, in the corrected version
- Lines 628, 638: D2O, please remove “2H”
Correction done in Line 696, in the corrected version
Reviewer 3 Report
The manuscript describes the production of exopolysaccharides by Ramlibacter tataouinensis and characterization of EPSs using different techniques. The approach is straightforward, mostly well described, and the text is typically easy to follow. I recommend the manuscript for publication after minor revision:
-The name of bacterium in the title should be italic.
- The OD in batch cultivation was 0.16. However, the cell dry weight was 32 g/L (4.8 mg from 150 mL in line 109). How is it possible that very low OD can make such this cell dry weight?
- Line 191. Please use subscript for 2 and 4 for H2SO4.
Author Response
The manuscript describes the production of exopolysaccharides by Ramlibacter tataouinensis and characterization of EPSs using different techniques. The approach is straightforward, mostly well described, and the text is typically easy to follow. I recommend the manuscript for publication after minor revision:
-The name of bacterium in the title should be italic.
Modification done.
- The OD in batch cultivation was 0.16. However, the cell dry weight was 32 g/L (4.8 mg from 150 mL in line 109). How is it possible that very low OD can make such this cell dry weight
4.8 mg in 150 mL : 32 mg/L not 32 g/L
- Line 191. Please use subscript for 2 and 4 for H2SO4.
Modification done.
Round 2
Reviewer 2 Report
Dear Authors,
The corrected version is definitely more legible and transparent. Despite still "NMR spectrometry" is present and suplementary file named "biosyntesis..."
I agree, the manuscript posses many data, however preliminary structure of EPS is presented. For presentation of the complete structure of EPS you need experiments such as NOESY/HMBC - which will give a proof for EPS sequence. The presented data in table S7 are not complete (without describing the spin systems, identification is impossible) and have shown huge heterogeneity of sample.
The present manuscript could be accepted with the minor revision as an Article to Molecules.
Author Response
Reviewer 2 (2)
The corrected version is definitely more legible and transparent. Despite still "NMR spectrometry"
Modification done
is present and suplementary file named "biosyntesis..."
Modification done
I agree, the manuscript posses many data, however preliminary structure of EPS is presented. For presentation of the complete structure of EPS you need experiments such as NOESY/HMBC - which will give a proof for EPS sequence.
The presented data in table S7 are not complete (without describing the spin systems, identification is impossible) and have shown huge heterogeneity of sample.
We did not perform NOESY/HMBC spectra; we had already used 240 scans per increment for HSQC NMR spectra, and HMQC NMR spectra with significant cross peaks require about 10 times the number of scans like HSQC NMR spectra. Owing to mixture complexity of EPSs as seen e.g .from HSQC NMR spectra, HMBC spectra would have been extremely crowded and probably not led to further meaningful cross peaks (note that common HMBC NMR spectra show fine structure in 1H dimension, further contributing to superposition of cross peaks). Total acquisition time would have been prohibitively long as high resolution in 13C dimension would have been necessary in HMBC NMR spectra because all carbohydrates share common NMR characteristics resonating within a small overall range of chemical shift. Extensive superposition of cross peaks is also anticipated for NOESY NMR spectra, as seen by highly complex TOCSY NMR spectra.
We achieved to purify the EPS in a reproducible way, but we know that there is a heterogeneity of molecules present as well with the EPS, e.g. lipids, and also different oligomers of this EPS, which complicates the NMR spectra. We did not hide this information and we clearly mentioned it in the text and in our first response. The structure that we proposed (not asserted) is based on the whole set of analyses we performed, mainly FT-ICR-MS and GC-MS, then NMR, FT-IR and elemental analysis. Thus, our purpose was not to assign all the signals from the NMR spectra, but to seek for expected signals based on the FT-ICR-MS fragments we identified. The signals that we assigned are clearly stated in Figure S9. Thus, we cannot assign all the spin systems in Table S7. In order to clarify these facts, we have added a comment in the legend of Table S7.
One should be aware that the characterization of very complex EPS, strongly bound to the cyst membrane is rarely described, except for macromolecules of very simple structure (like alginate in the case of Azotobacter).
Our paper relies on a methodology based on a pool of analytical techniques. As mentioned in the article, the final structure of the EPS is proposed, it is not confirmed, but it is a first step to address this type of complex task. This proposal can move the scientific community forward, and make people think about the use of FT-ICR-MS as a tool in this kind of technological challenge.
The present manuscript could be accepted with the minor revision as an Article to Molecules.